

# Unconventional superconductivity in a strongly correlated band-insulator without doping

**Anwesha Chattopadhyay[1], H. R. Krishnamurthy [2] and Arti Garg[1]⋆**

**1** Condensed Matter Physics Division, Saha Institute of Nuclear
Physics, HBNI, 1/AF Bidhannagar, Kolkata 700 064, India
**2** Centre for Condensed Matter Theory, Department of Physics,
Indian Institute of Science, Bangalore 560 012, India

⋆ arti.garg@saha.ac.in

## Abstract

We present a novel route for attaining unconventional superconductivity in a strongly correlated system without doping. In a simple model of a correlated band insulator at half-filling we demonstrate, based on a generalization of the projected wavefunctions method, that superconductivity emerges for a broad range of model parameters when e-e interactions and the bare band-gap are both much larger than the kinetic energy, provided the system has sufficient frustration against the magnetic order. As the interactions are tuned, the superconducting phase appears sandwiched between the correlated band insulator followed by a paramagnetic metal on one side, and a ferrimagnetic metal, antiferromagnetic half-metal, and Mott insulator phases on the other side.

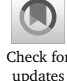
# 1   Introduction

The discovery of unconventional superconductivity in a variety of materials, such as high $T_c$ superconductivity in cuprates [1], iron pnictides and chalcogenides [2], in organic superconductors [3], in heavy fermions [4] and very recently in magic angle twisted bilayer graphene [5,6], has always ignited worldwide interest owing to their rich phenomenonology, the theoretical challenges they pose, scientific implications and broad application potential.  In almost all of these examples, superconductivity appears upon chemically doping the parent compound away from commensurate filling [1, 2, 5–8], though in some cases inducing charge fluctuations by changing pressure also leads to the superconducting phase [3, 8].  An important experimental fact is that chemical doping inevitably induces disorder, as is clearly the case in high $T_c$ superconductors (SCs), which makes these materials very inhomogeneous [9–12].  It is a theoretical and experimental challenge to come up with new mechanisms and materials for clean high $T_c$ SCs.

Theoretical analysis has shown that strong e-e correlations are crucial to achieve unconventional superconductivity.  In most of the known unconventional SCs [1–3, 5–8] the low temperature phase of the parent compound is either a strongly correlated AF Mott insulator where charge dynamics is completely frozen, or a AF spin-density-wave phase with at least moderately strong correlations.  The unconventional superconductivity in many of these materials can be understood, at least qualitatively, in terms of the strongly correlated limit of the paradigmatic Hubbard model (single or multi band) doped away from half-filling [7,8,13–16].  But the possibility of a SC phase in a *strongly correlated band-insulator* has been explored very little so far, either theoretically or experimentally.

In this work, we show how a spin-exchange mediated SC can be realized *without doping* in a simple model of a strongly correlated band insulator (BI), where the bare band gap and the e-e interactions both dominate over the kinetic energy.  As e-e interactions are increased (but still remain of the order of the band-gap), the single particle excitation gap in the BI closes, resulting in a metallic phase.  Upon further increasing the e-e interactions, superconductivity develops by the formation of a coherent macroscopic quantum condensation of electron pairs, provided the metal has enough low energy quasiparticles and the system has enough frustration against the magnetic order.  The superconductivity, which survives for a broad range of e-e interactions, features tightly bound short coherence length Cooper pairs with a $T_c$ well separated from the energy scale at which the pairing amplitude builds up.  The phase diagram, whose section with all model parameters fixed except for the interaction to band-gap ratio is shown in Fig. 1, presents a plethora of exotic phases, that we discuss further below, in the vicinity of a broad region of the SC phase.

# 2   Ionic Hubbard model and the limit of strong correlations

Our starting point is a variant of the Hubbard model, known as the *ionic* Hubbard model (IHM), where, on a bipartite lattice with sub-lattices A and B, a staggered ionic potential $\Delta/2$

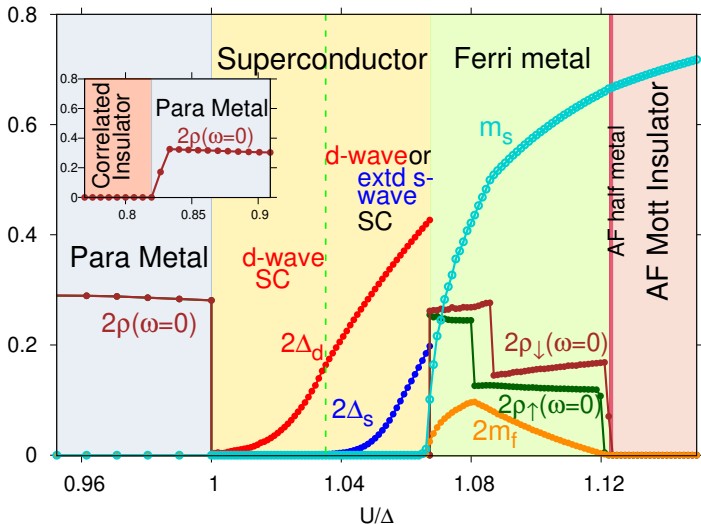

Figure 1: **Phase Diagram at a fixed** $t'$ . The zero temperature phase diagram for the 2d square lattice for $U = 10t$ and $t' = 0.4t$. For $\Delta \gg U \gg t$, the system is a correlated band insulator without any magnetic order which is adiabatically connected to the BI at $U = 0$. On increasing $U$, first the gap in the single particle excitation spectrum closes, as shown by the non-zero single particle density of states (DOS) at the Fermi energy $\rho(\omega = 0)$, resulting in a metallic phase. On further increasing $U/\Delta$, superconductivity sets in and lasts over a broad range ($\Delta \in [9.3 : 10]t$) before the ferrimagnetic order with a non-zero staggered magnetization ($m_s$) and non zero uniform magnetization ($m_f$) sets in via a first order transition. This is a Ferri metal phase with $\rho_\uparrow(\omega = 0) \neq \rho_\downarrow(\omega = 0) > 0$. As $U/\Delta$ increases further, $m_f \to 0$ whence the magnetic order becomes AF. Furthermore, a spectral gap opens up for the up-spin electrons such that $\rho_\uparrow(\omega = 0) = 0$ while the down-spin electrons are still conducting with $\rho_\downarrow(\omega = 0)$ being finite, resulting in a sliver of AF half-metal. Eventually the system becomes a AF Mott insulator as $U/\Delta$ increases further. *Note that the SC phase is surrounded by metallic phases on both the sides*.

is present in addition to electron hopping and coulomb repulsion ($U$):

$$\mathcal{H} = -\sum_{i,j\sigma}(t_{ij}c_{i\sigma}^\dagger c_{j\sigma} + h.c.) - \mu\sum_i n_i$$
$$- \frac{\Delta}{2}\sum_{i\in A}n_i + \frac{\Delta}{2}\sum_{i\in B}n_i + U\sum_i n_{i\uparrow}n_{i\downarrow}. \tag{1}$$

The amplitude for electrons with spin $\sigma$ to hop between sites $i$ and $j$ is $t_{ij} = t$ for near-neighbours and $t_{ij} = t'$ for second neighbours. The chemical potential $\mu$ is chosen to fix the average site occupancy at $n = 1$, corresponding to half-filling. The staggered potential doubles the unit cell, and (for $t' < \Delta/4$) induces a gap between the two electronic bands that result, making the system a BI at half-filling when the Hubbard on-site interaction $U$ is zero. The Hamiltonian in Eq. 1 under the particle-hole transformation $c_{iA\sigma}^\dagger \to -c_{iB\sigma}$ and $c_{iB\sigma}^\dagger \to c_{iA\sigma}$ maps to $H(-t')$. Thus, in this paper we restrict our attention to positive values of $t'$. The physics for negative values of $t'$ is obtainable simply via the particle-hole transformation.

The parameter range of interest for this work is $U \sim \Delta \gg t, t'$, where a theoretical solution can be obtained based on a generalization of the projected wavefunctions method [13,17–23]. In this limit and at half-filling, holons are energetically expensive on the $A$ sites (with onsite

potential $-\frac{\Delta}{2}$) and doublons are expensive on the $B$ sites (with onsite potential $\frac{\Delta}{2}$); i.e., in the low energy subspace $h_A$ and $d_B$ are constrained to be zero (with $d$ representing a doublon and $h$ a holon). Consequently, we can carry out a similarity transformation to eliminate all hopping processes connecting the low and high energy sectors of the Hilbert space. Nevertheless, and unlike in the Hubbard model, in the half-filled IHM the system still has charge dynamics through hopping processes which take place entirely within the low-energy Hilbert space, e.g., first neighbor processes such as $|d_A h_B\rangle \Longleftrightarrow |\uparrow_A \downarrow_B\rangle$ and second neighbour hopping processes which allow doublons to hop on the A sublattice and holons to hop on the B sublattice. Further details can be found in Appendix A.

The effective low energy Hamiltonian at half-filling, $H_{eff}$, is an extended $t - t' - J - J'$ model acting on a projected Hilbert space:

$$
\begin{aligned}
H_{eff} = &-t \sum_{<ij>,\sigma} \mathcal{P}[c^\dagger_{iA\sigma} c_{jB\sigma} + h.c.]\mathcal{P} - t' \sum_{<<ij>>,\alpha,\sigma} \mathcal{P}[c^\dagger_{i\alpha\sigma} c_{j\alpha\sigma} + h.c.]\mathcal{P} \\
&+ J' \sum_{<<ij>>} \mathcal{P}\left[S_{iA}.S_{jA} - \frac{1}{4}(2-n_{iA})(2-n_{jA})\right] + \left[S_{iB}.S_{jB} - \frac{1}{4}n_{iB}n_{jB}\right]\mathcal{P} \\
&+ J \sum_{<ij>} \mathcal{P}(S_{iA}.S_{jB} - (2-n_{iA})n_{jB}/4)\mathcal{P} + H_0 + H_d + H_{tr} - \mu \sum_i n_i + ...
\end{aligned} \tag{2}
$$

Here $J = 2t^2/(U+\Delta)$ and $J' = 4t'^2/U$. $H_0$ is the rescaled Hubbard interaction term in the projected Hilbert space. $H_d(H_{tr})$ indicates other dimer (trimer) processes. We treat the projection constraint in $H_{eff}$ using the generalised Gutzwiller approximation [22] and solve it using a renormalized Bogoliubov mean field theory. Gutzwiller approximations [19, 20, 22] of the sort we use have been well vetted against quantum Monte Carlo calculations [13, 18, 23] and dynamical mean field theory [24]. Details of the Gutzwiller approximation and the various terms in $H_{eff}$ are given in Appendix A.

## 3 Phase diagram and the order parameters

We solve the renormalized effective low energy Hamiltonian using a renormalized mean field theory (RMFT), in which the spin-exchange terms and various dimer and trimer terms are decomposed into the quadratic form both in the particle-particle and the particle-hole channels giving non-zero expectation value to the following mean fields: (a) pairing amplitude, $\Delta^\gamma_{AB} \equiv \langle c^\dagger_{iA\uparrow} c^\dagger_{i+\gamma B\downarrow} - c^\dagger_{iA\downarrow} c^\dagger_{i+\gamma B\uparrow}\rangle$, where $\gamma$ is $x$ or $y$, considering d-wave pairing symmetry ($\Delta^x_{AB} = -\Delta^y_{AB} \equiv \Delta_d$) and extended s-wave pairing symmetry ($\Delta^x_{AB} = \Delta^y_{AB} \equiv \Delta_s$) separately; (b) density difference between two sublattices, $\delta = (n_A - n_B)/2$; (c) inter sublattice fock shifts, $\chi^{(1)}_{AB\sigma} = \langle c^\dagger_{iA\sigma} c_{jB\sigma}\rangle, j = i \pm x, i \pm y$, $\chi^{(2)}_{AB\sigma} = \langle c^\dagger_{iA\sigma} c_{jB\sigma}\rangle, j = i \pm 2x \pm y$ or $i \pm 2y \pm x$; (d) intra sublattice fock shift on A(B) sublattice, with $\chi_{\alpha\alpha\sigma} = \langle c^\dagger_{i\alpha\sigma} c_{i\pm 2x/2y\alpha\sigma} + h.c.\rangle$, and $\chi'_{\alpha\alpha\sigma} = \langle c^\dagger_{i\alpha\sigma} c_{i\pm x\pm y\alpha\sigma} + h.c.\rangle$; (e) and the magnetic order parameters, namely, the staggered magnetization $m_s = (m_A - m_B)/2$ and the uniform magnetisation $m_f = (m_A + m_B)/2$, where $m_{A,B}$ is the sublattice magnetization. We have kept the contribution of nearest neighbour spin-exchange term, dimer ($H_d$) and trimer ($H_{tr}$) terms in the pairing amplitudes in order to study the d-wave and the extended s-wave SC for which Cooper pairs live on the nearest-neighbour bonds, though other mean fields mentioned above have contribution from the second neighbour spin-exchange term as well. Details of the mean field Hamiltonian, and self consistent equations for pairing gap and magnetic order parameters are given in Appendix B.

We basically study three different versions of the renormalized mean field theory (RMFT). (1) To explore the SC phase, we use a generalised spin-symmetric Bogoliubov mean field theory (i.e., with the magnetic order parameters set to zero), which basically maps onto a two-

site Bogoliubov-deGennes (BdG) mean field theory for each allowed $k$ point in the BZ. (2) To explore the magnetic order and the phase transitions involved, we solve the renormalized Hamiltonian using standard mean field theory allowing non-zero values of the sublattice magnetization $m_\alpha = n_{\alpha\uparrow} - n_{\alpha\downarrow}$ with $\alpha = A, B$, along with all other mean-fields mentioned above except for the SC pairing amplitudes $\Delta_{s/d}$. (3) The third calculation, where we allow for both the SC pairing amplitudes and the magnetization along with all other mean fields metioned above, uses a standard canonical transformation followed up by the Bogoliubov transformation to diagonalise the mean field Hamiltonian neglecting the inter-band pairing as weak. We solve the resulting RMFT self-consistent equations on the square lattice for various values of $U, \Delta$ and $t'$ to obtain the phase diagram shown in Fig. 1 and Fig. 2 (See Appendix C for details). In the parameter regime where solutions with nonzero SC pairing amplitudes and magnetization (from the first two calculations) are both viable, we compare the ground state energy of the two mean-field solutions to determine the stabler ground state. We finally compare the energy of this state with the one obtained in the third calculation to determine the true ground state.

Our main findings are summarised in the phase diagram of Fig. 1, which shows a linear section (along the $U/\Delta$ axis) of the full phase diagram in Fig. 2[e], for the IHM on a 2d square lattice. The unconventional SC phase is sandwiched between paramagnetic and ferrimagnetic metallic phases, which in turn are sandwiched between a correlated band insulator and an AF Mott insulator (MI), along with an intervening sliver of AF half-metal. The correlated band insulator, stable for $\Delta \gg U \gg t$, is paramagnetic and adiabatically connected to the BI phase of the non-interacting IHM. As $\Delta$ approaches $U$, the low energy hopping processes ($|d_A h_B\rangle \Longleftrightarrow |\uparrow_A \downarrow_B\rangle$) become more prominent, increasing charge-fluctuations such that the gap in the single particle excitation spectrum closes, leading to a finite density of states (DOS) $\rho(\omega = 0)$ at the Fermi energy, though for most of the parameter regime the resulting paramagnetic metallic (PM) phase is a compensated semi-metal with small Fermi pockets as shown in detail in Fig. 3. This PM phase is adiabatically connected to the metallic phase observed for weak to intermediate strength of $U/t$ as long as $U \sim \Delta$ and the system is constrained to be paramagnetic, as shown in earlier work on the IHM using DMFT and other approaches [25–28]. On further increasing $U/\Delta$, in the presence of sufficiently large $t'$, superconductivity sets in for $U \sim \Delta$ (irrespective of the strength of $U/t$, as shown in Appendix D) due to the formation of coherent Cooper pairs of quasi-particles which live near the Fermi pockets, and survives for a broad range of $U/\Delta$.

The pairing amplitude $\Delta_{d/s}$ for both the pairing symmetries we have studied, namely, the d-wave and the extended s-wave, increases monotonically with $U/\Delta$ and drops to zero via a first order transition at the transition to the ferrimagnetic metal. Though there is a metastable state in which the SC phase coexists along with the ferrimagnetic order for a range of $U/\Delta$ after the transition (see Appendix C for details), due to the really tiny Zeeman splitting ($\leq 0.035t$ for $U = 10t$) produced by the small uniform magnetization $m_f$ the possibility of a Fulde-Ferrel-Larkin-Ovchinnikov (FFLO) state seems unlikely [29–31].

The ferrimagnetic metal (FM) phase is characterised by non-zero values of the staggered magnetization $m_s$ as well as the uniform magnetization $m_f$, along with a finite DOS $\rho_\sigma(\omega = 0)$ at the Fermi energy. With further increase in $U/\Delta$ the FM evolves into an AF *half-metal* phase in which the system has only staggered magnetization (i.e., $m_f = 0$) and the single particle excitation spectrum for up-spin electrons is gapped while the down-spin electrons are still in a semi- metal phase. Eventually, for a large enough $U/\Delta$, both the spin spectra become gapped, and the system becomes an AF MI. Though we have studied the IHM on a square lattice, a qualitatively similar phase diagram is expected on any bipartite lattice, but with changes involving appropriate symmetries, e.g., $d + id$ pairing symmetry on a honeycomb lattice. We would also like to emphasize that though most of the results presented in this work are for a

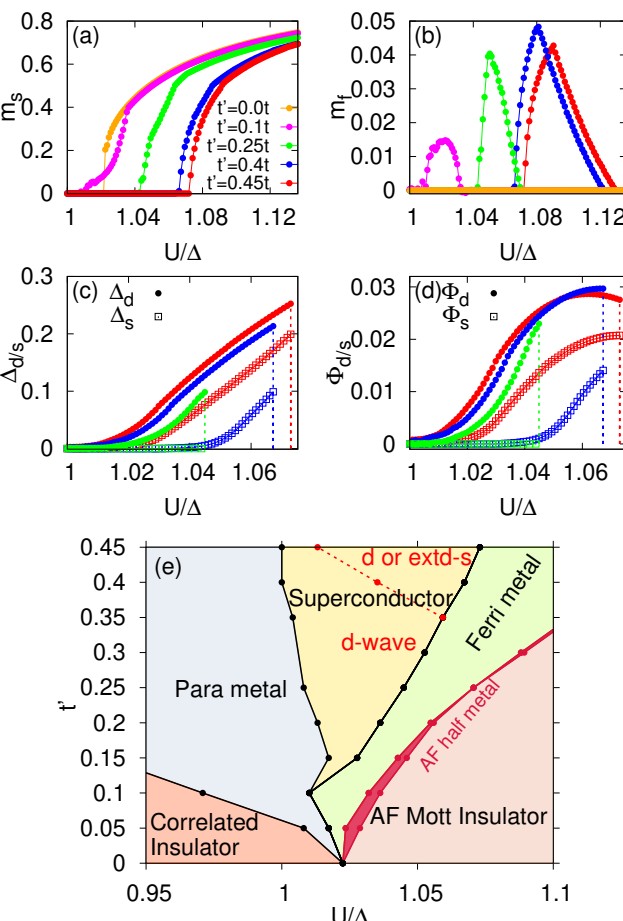

Figure 2: **Order Parameters and Complete Phase diagram.** Top panels show the staggered magnetization, $m_s$ and the uniform magnetization, $m_f$ as functions of $U/\Delta$ for several values of $t'$ and $U = 10t$. With increasing $t'$, the transition point at which the magnetic order turns on first decreases for $t' \leq 0.12$ and then starts increasing again. The magnetic transition is of first order for $t' = 0$ as well as for large values of $t'$, though for intermediate values of $t'$ the magnetization tuns on continuously. Panel (c) shows the SC pairing amplitude $\Delta_{d/s}$, for the d-wave and extended s-wave pairing symmetry. With increasing $t'$ the range in $U/\Delta$ over which the superconductivity is stable gets wider, and the amplitudes of both d-wave and extended s-wave pairings get enhanced. Note that the extended s-wave order turns on only for $t' > 0.35t$. Panel (d) shows the SC order parameter $\Phi_{d/s}$, which also gives an estimate of the SC transition temperature, $T_c$. The bottom panel (e) shows the complete zero temperature phase diagram for $U = 10t$ in the $t'$-$U/\Delta$ plane. As we approach the SC phase from either the correlated band insualtor or the MI phase, the charge fluctuations build up gradually through metallic phases, and the superconductivity develops by the formation of coherent Cooper pairs between electrons which reside on the Fermi pockets of these metallic phases.

2d square lattice, the phase diagram obtained within the renormalized mean field theory for higher dimensional lattices is qualitatively similar, as seen in the phase diagram for a 3d cubic lattice shown in Appendix D.

We next discuss the changes in behavior of the system with increasing $U/\Delta$ for varying values of $t'$, as depicted in Fig. 2. For $t' = 0$, the system shows a direct first order transition from an AF ordered phase to a correlated band insulator with a sliver of a half-metallic AF phase

close to the AF transition point. This is consistent with a variational quantum Monte Carlo study of the half-filled IHM for $t' = 0$ [32] as well as with most other earlier work [33, 34]. When $t'$ is non-zero, due to the breaking of particle-hole symmetry as well as the frustration induced by the second neighbour spin-exchange coupling $J'$, the system first attains ferrimagnetic order characterized by non-zero values of both the staggered ($m_s$) and the uniform ($m_f$) magnetizations, for a range of $U/\Delta$, beyond which it has pure AF order as shown in panel (a) of Fig. 2. The magnetic transition occurs at increasingly larger values of $U/\Delta$ with increasing $t'$ (except for an initial decrease for small values of $t'$) which helps in the development of a stable SC phase.

To stabilize the superconducting phase, a minimum threshold value of $t'$ (which is a function of $U$) is required, partly in order to frustrate the magnetic order as mentioned above, but more importantly to gain sufficient kinetic energy by intra-sublattice hopping of holons and doublons on their respective sublattices where they are energetically allowed. While a stable d-wave SC phase turns on for $t' > 0.1t$ for $U = 10t$, as shown in Fig. 2 superconductivity in the extended s-wave channel gets stabilized for the much larger value of $t' > 0.35t$ . In an intermediate regime of $U/\Delta$ and $t'$, states with both d-wave and extended s-wave symmetry are viable solutions with energies that are very close (See Appendix C for details). As $t'$ increases, the pairing amplitude increases and the range of $U/\Delta$ over which the SC phase exists becomes broader for both the pairing symmetries studied. Though $t'$ helps in the formation of the SC phase with pairing amplitudes living on the nearest neighbour bonds, there is no significant second neighbour pairing induced by $J'$.

The pairing amplitude discussed above signals the strength of Cooper pairing on a bond, but the SC order parameter $\Phi_{d/s}$ is defined in terms of the off-diagonal long-range order in the correlation function $F_{\gamma_1\gamma_2}(\mathbf{r}_i - \mathbf{r}_j) = \langle B^\dagger_{i\gamma_1} B_{j\gamma_2} \rangle$ where $B^\dagger_{i\gamma}$ creates a singlet on the bond $(i, i+\gamma)$. Fig. 2 shows the SC order parameter, which has been obtained after taking care of renormalization required in $F_{\gamma_1\gamma_2}(\mathbf{r}_i - \mathbf{r}_j)$ in the projected wavefunction scheme. Since the SC order parameter for this system is much smaller than the strength of the pairing amplitude, with increase in temperature the superconductivity will be destroyed at $T_c$ by the loss of coherence among the Cooper pairs, leaving behind a pseudo-gap phase with a soft gap in the single particle density of states due to the Cooper pairs which will exist even for $T > T_c$. Thus $\Phi_{d/s}$ also provides an estimate of the SC transition temperature $T_c$. The maximum estimated $T_c$ for $U = 10t$ on a square lattice is approximately $0.03t$ for the d-wave SC phase. For a hopping amplitude $t \sim 0.4eV$ (which happens to be the scale for cuprates), our estimated transition temperature is $T_c \sim 150K$, and there is a considerable scope for enhancing $T_c$ by tuning $U/\Delta$ as well as $t'$.

We note that, in an earlier work [35] on the strongly correlated half-filled IHM with $t' = 0$, (i.e., in the absence of any of the frustration effects we have discussed above,) using slave bosons to represent the projection processes in Eq. 2, and using a slave-boson mean field theory approach to treat the problem, SC was shown to exist when $U \sim \Delta \gg t$. However, this result is not consistent with the variational quantum Monte-Carlo study mentioned above [32] where no SC phase was reported at half-filling in the absence of frustration against the magnetic order. Within the Gutzwiller projection approach, while we do find regions of parameter space inside the AFI region where SC pairing is viable even in the $t' = 0$ case, the SC phase has higher energy than the AFI phase and is therefore metastable [24]; and as we have demonstrated above, only in the presence of sufficient frustration against the magnetic order does SC exist in this simple model of a band-insulator at *half-filling*.

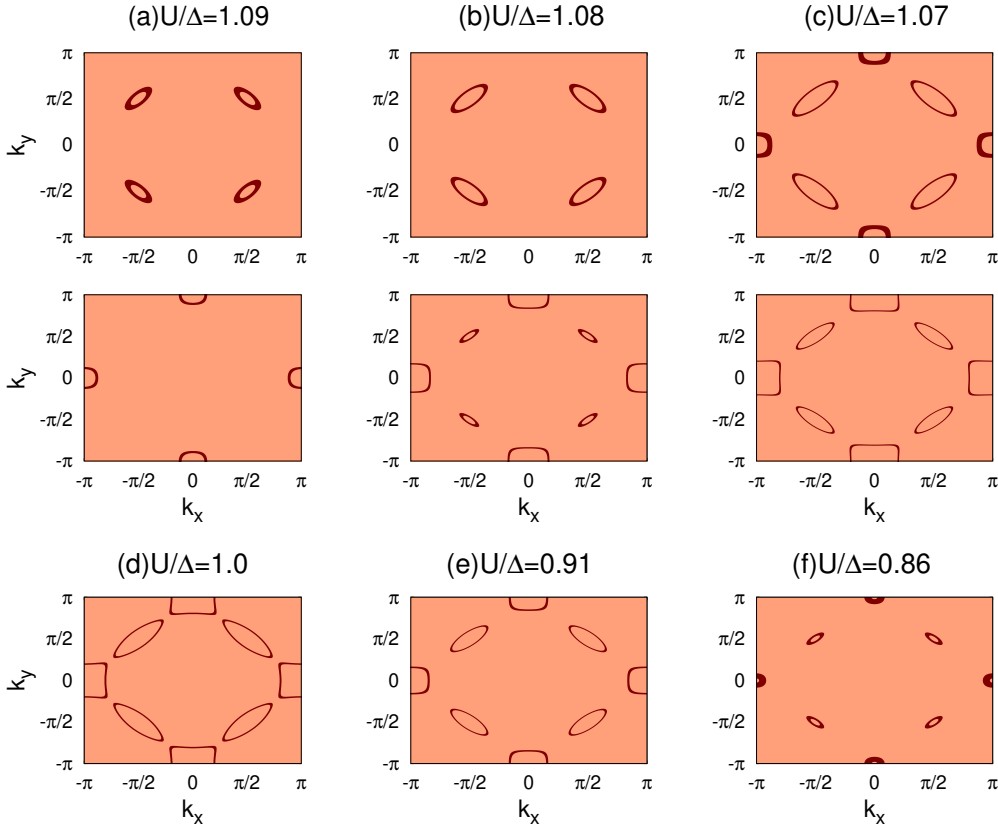

Figure 3: **Spectral Functions.** The top two rows show the spin resolved low energy spectral functions $A_\sigma(k, \omega \sim 0)$ (integrated over $|\omega| \leq (0.01 - 0.02)t$ for a $3000 \times 3000$ system) in the full Brillouin Zone (BZ) for $t' = 0.35t, U = 10t$, to emphasize how the charge fluctuations evolve as we approach the SC regime from the ferri metal side, with $A_\uparrow(k, \omega \sim 0)(A_\downarrow(k, \omega \sim 0))$ shown in the first (second) row. At $U/\Delta = 1.09$, the up spin channel has electron pockets while the down spin channel has small hole pockets. As $U/\Delta$ decreases, these Fermi pockets become bigger, the down spin spectral function gets additional electron pockets and the up-spin spectral functions get additional hole pockets. The last row shows $A(k, \omega \sim 0)$ (same for up or down spins) for the para metal phase. Moving towards the SC phase by increasing $U/\Delta$, Fermi pockets in the para metallic state go on expanding until they almost start touching each other, at which point the superconductivity sets in by formation of Cooper pairs between electrons close to the Fermi energy.

## 4 Spectral functions and single-particle density of states

A striking feature of the phase diagram in Fig. 2 is that, though the origin of superconductivity in this model lies predominantly in the spin-exchange interactions (with a weaker contribution from other dimer and trimer terms), superconductivity sets in only after the system has evolved to a para metallic or a FM phase. In order to understand the charge dynamics as the system approaches the SC phase with the tuning of $U/\Delta$, we have analysed the single particle spectral functions which can be directly measured in angle resolved photoemission spectroscopy (ARPES). Fig. 3 shows the low energy spin resolved spectral functions $A_\sigma(k, \omega \sim 0)$ averaged

over the two sublattices, the non-zero value of which determine the energy contour on which low energy quasiparticles live in the Brillouin zone (BZ) (see Appendix B for details). Panels (a-c) show $A_\sigma(k, w \sim 0)$ in the FM phase for which the up-spin channel has electron pockets around the points $\mathbf{K} = (\pm\pi/2, \pm\pi/2)$ in the BZ and the down spin spectrum has small hole pockets around the points $\mathbf{K}' = (\pm\pi, 0), (0, \pm\pi)$ in the BZ as shown in panel (a). As $U/\Delta$ decreases within the FM phase, and approaches the SC phase, the electron pockets (hole-pockets) in the up-spin (down-spin) spectral function become bigger, the down-spin channel gets additional electron pockets while the up-spin channel gets additional hole pockets as shown in panel (c). In the PM phase, the low energy spectral functions have both electron pockets

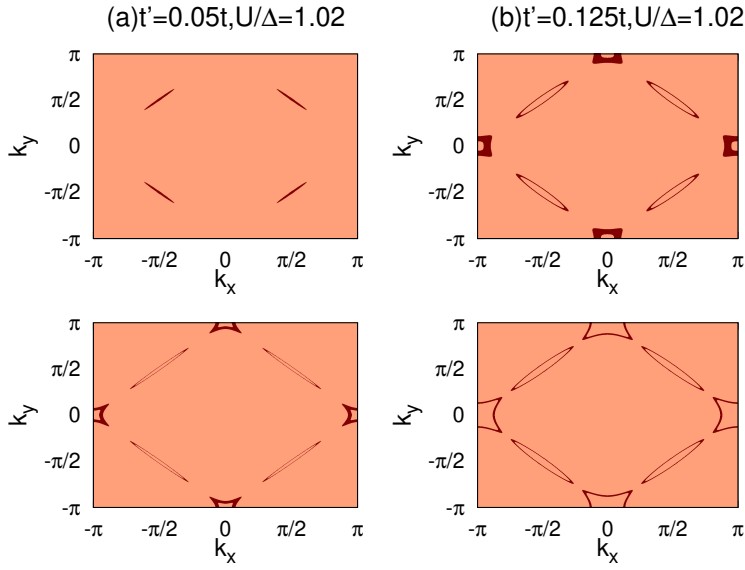

Figure 4: **Spectral functions**. Here we show the low energy spectral functions $A_\sigma(k, \omega \sim 0)$ (integrated over $|\omega| \le (0.01 - 0.02)t$ on a $3000 \times 3000$ lattice) in the full Brillouin zone (BZ) for the ferrimagnetic phase at a fixed $U/\Delta = 1.02$ and for two values of $t'$. Upper panels show $A_\uparrow(k, \omega \sim 0)$, and the bottom panels $A_\downarrow(k, \omega \sim 0)$.

(around $\mathbf{K}$) as well as the hole pockets (around $\mathbf{K}'$). As $U/\Delta$ increases through the PM phase, these Fermi pockets slowly expand such that they almost touch each other before the system enters into the SC phase. Similar behaviour is seen with an increase of $t'$ in the PM or the FM phases. We note that the quasiparticle weight in the sublattice-averaged spectral function, which is the one of experimental relevance and the one shown in Fig. 3, is essentially constant throughout the Fermi pockets in the para metal phase and has a very weak variation over the Fermi pockets in the ferri-metal phase (see Appendix B for details). However, the spectral function for each sublattice evaluated separately does show a variation in the quasiparticle weight over the Fermi pockets.

In order to understand the charge dynamics as the system approaches the SC phase with the tuning of second neighbour hopping, $t'$, we have analysed the single particle spectral functions for a fixed $U/\Delta$ in the ferrimagnetic metallic phase. We can understand why the SC phase does not get stabilized for small values of $t'$ by looking at the evolution of $A_\sigma(k, \omega \sim 0)$ for a fixed $U/\Delta$ as one tunes $t'$. Fig. 4 shows $A_\sigma(k, \omega \sim 0)$ close to the magnetic transition point of $t' = 0$, that is, for $U/\Delta = 1.02$. For small values of $t'$, at this value of $U/\Delta$ the system is in the ferrimagnetic metal phase. As we increase $t'$ inside the ferrimagnetic metal phase, the up spin spectral functions get bigger electron pockets around $\mathbf{K} = (\pm\pi/2, \pm\pi/2)$ points

while the down spin spectral functions get bigger hole pockets around $\mathbf{K}' = (\pm\pi, 0), (0, \pm\pi)$ points. In addition to this, as $t'$ increases even the up-spin spectral functions get hole pockets and the down spin spectral functions get electron pockets. As a result of both these effects, an almost connected contour of Fermi pockets is formed, whence superconductivity emerges by the formation of Cooper pairs of the corresponding low energy quasiparticles.

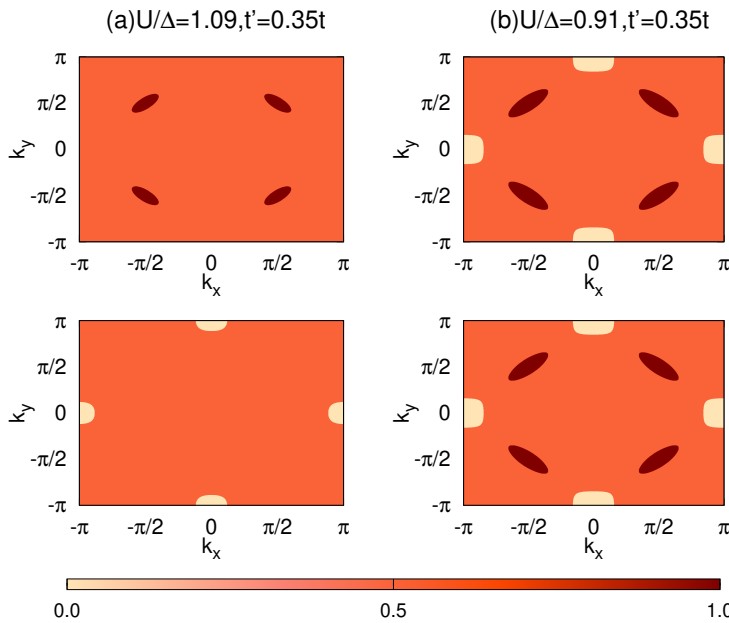

Figure 5: **Momentum Distribution Function**. Momentum distribution function $n_\sigma(k)$ in the ferrimagnetic metal and the para metal phases for $t' = 0.35t$. In the ferrimagnetic metal phase shown in panel (a) $n_\uparrow(k) > 1/2$ on (electron) pockets centered around the $\mathbf{K}$ points while $n_\downarrow(k) < 1/2$ on (hole) pockets centered around the $\mathbf{K}'$ points in the BZ. Panel (b) shows the results for the paramagnetic metal phase, where the systen has spin symmetry and $n_\sigma(k) < 1/2$ around the $\mathbf{K}'$ points while $n_\sigma(k) > 1/2$ around the $\mathbf{K}$ points for both the spin components. Everywhere else in the BZ $n_\sigma(k) = 1/2$ in all the panels.

The electron and hole pockets mentioned above are best identified based on the momentum distribution function $n_\sigma(k)$ as defined in Appendix A. $n_\sigma(k)$ is uniformly half in the entire BZ for any insulating phase of the model studied here. When the system goes into a metallic phase, at least one of the bands cross the Fermi level resulting in filled or empty Fermi pockets depending on the curvature of the band. Filled Fermi pockets, also called electron pockets, have $n_\sigma(k) > 1/2$, while empty Fermi pockets, also called hole pockets, have $n_\sigma(k) < 1/2$. Fig. 5 shows $n_\sigma(k)$ for $t' = 0.35t$ for two values of $U/\Delta$. Panel (a) shows the result for the ferrimagnetic metal phase and panel (b) shows the results in the para metal phase. In the ferri-metal phase, $n_\uparrow(k)$ has filled pockets around the $\mathbf{K}$ points while the down-spin component has hole pockets around the $\mathbf{K}'$ points in the BZ. In the para-metal phase, shown in panel (b), there is a spin symmetry and $n_\sigma(k)$ has electron and hole pockets for both the spin channels.

Fig. 6. shows the spin-resolved single particle density of states (DOS) $\rho_\sigma(\omega)$ which can be measured directly in scanning tunneling spectroscopy (STS) experiments and provides additional evidence for the existence of various metallic phases as in the phase diagram in Fig. 2. The DOS at $\omega = 0$ for these phases was presented in Fig. 1 as a function of $U/\Delta$, and here we present the full $\rho_\sigma(\omega)$ vs $\omega$. The para metal, ferri-metal and the AF half-metal phases are all compensated semi metals, which is reflected in the depletion in the DOS at the Fermi energy

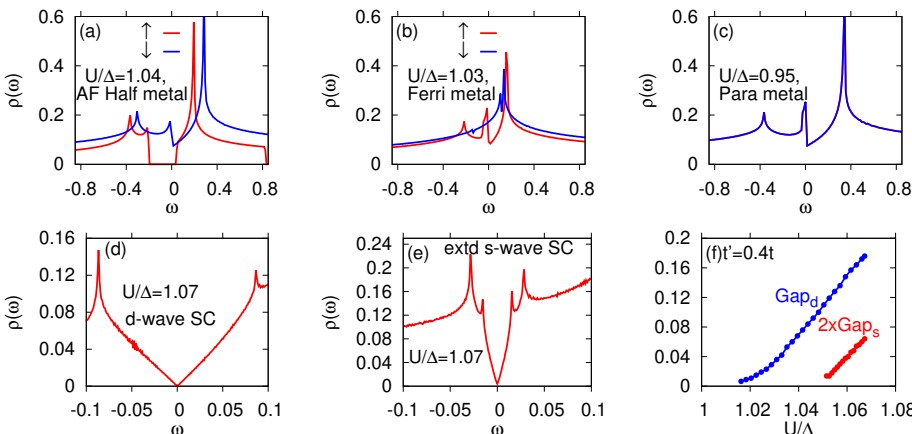

Figure 6: **Single particle Density of states.** Panels (a)-(c) show the spin resolved single particle density of states (DOS) $\rho_\sigma(\omega)$ for $t' = 0.15t$ and $U = 10t$. At $U/\Delta \sim 1.04$, $\rho_\downarrow(\omega = 0)$ is finite where as $\rho_\uparrow(\omega = 0) = 0$ with a finite spectral gap, corresponding to the AF half-metal phase. At $U/\Delta = 1.03$, the DOS at the Fermi energy is finite in both the spin channels but $\rho_\uparrow(\omega) \neq \rho_\downarrow(\omega)$ corresponding to the ferri metal phase. At $U/\Delta = 0.95$, the DOS is spin symmetric with a finite weight $\rho_\sigma(\omega = 0)$ at the Fermi energy and the system is a para metal. Panel (d) shows $\rho(\omega)$ for the d-wave SC phase while panel (e) shows that for the extended s-wave SC phase for $U = 10t$ and $t' = 0.4t$. $\rho(\omega)$ shows a linear increase with $|\omega|$ for $\omega \sim 0$ for both the SC phases. Panel (f) shows the gap in the DOS, which is basically the peak to peak distance in $\rho_\sigma(\omega)$, for both the d-wave and the extended s-wave pairing symmetries.

and is consistent with the small Fermi pockets shown in Fig. 3 (See Appendix E for details). We have also analysed the DOS in the SC phase. As shown in Fig. 6[d], $\rho(\omega \sim 0) \sim |\omega|$ which is a signature of the gapless nodal excitations in the d-wave SC phase. Interestingly, even for the extended s-wave SC phase $\rho(\omega \sim 0) \sim |\omega|$ as the pairing takes place around small Fermi pockets which are centered at **K** or **K**′ points in the BZ where the pairing amplitude $\Delta_s(k) = \Delta_s(\cos(k_x) + \cos(k_y))$ has nodes as well, resulting in gapless excitations. The gap, which is the peak to peak distance in the DOS, is much larger in the d-wave SC phase than in the extended s-wave phase, consistent with the former being the stable phase. Infact for the extended s-wave phase, $Gap_s$ is only slightly larger than the SC order parameter $\Phi_s$, which indicates that the extended s-wave SC phase will have a narrower pseudogap phase above $T_c$, compared to the d-wave case.

# 5 Conclusions

As mentioned in the introduction, the origin as well as the basic features of unconventional SC in most of the superconducting materials known today [3, 5, 7, 8] can be understood, at least at the broad qualitative level [7, 8, 13–16], in terms of the strongly correlated limit of the Hubbard model (single or multi band), but only upon doping the system away from half-filling. In the theoretical model we have studied here, superconductivity appears even at half-filling, and therefore without the disorder that inevitably accompanies doping, in the special strongly correlated limit where $U, \Delta \gg t, t'$ and the second neighbour hopping is sufficiently strong. A remarkable feature is that the SC phase in this model of a correlated band insulator is sand-

wiched between paramagnetic metallic and ferrimagnetic metallic phases (Fig. 2[e]), which makes the zero temperature phase diagram very different from that of the known unconventional superconductors like high $T_c$ cuprates [7] or the more recent magic angle twisted bilayer graphene [5]. We expect that the SC phase in this model has transition temperatures comparable to those of cuprates and that it also has a pseudogap phase like in cuprates.

The question as to what are the possible experimental situations where this mechanism of superconductivity at half-filling, with its promise of large transition temperatures and no intrinsic disorder, can be realized is of obvious importance. Since the IHM has been realized for ultracold fermions on an optical honeycomb lattice [36], where the state-of-the art engineering allows the parameters in the Hamiltonian to be tuned with great control, it will be interesting and perhaps the easiest to explore our theoretical proposal in these systems. Due to the recent developments in layered materials and heterostructures, it is indeed possible to think of many scenarios where the IHM can be used as a minimal model, for example, graphene on h-BN substrate and bilayer graphene in the presence of a transverse electric field [37], which plays the role of the staggered potential. The limit of strong correlation, crucial for realizing the SC phase, can be achieved in these materials by applying a strain or twist. Band insulating systems with two inequivalent strongly correlated atoms per unit cell, frustration in hopping and antiferromagnetic exchange, and lack of particle-hole symmetry, are likely tantalizing candidate materials as well. Our work suggests that further theoretical and experimental exploration of such novel possibilities where superconductivity can be realized with sufficiently high transition temperatures without doping in strongly correlated band insulators is an exciting and worthwhile pursuit.

## Acknowledgements

We would like to acknowledge R. Sensarma for insightful discussions in early stages of this work. A. C. acknowledges financial support by Department of Atomic Energy, Government of India.

**Funding information**   A. G. and H. R. K gratefully thank the Science and Engineering Research Board of the Department of Science and Technology, India for financial support under grants No. CRG/2018/003269 and SB/DF/005/2017 respectively.

## A   Details of strong correlation limit and Gutzwiller projection

We first describe the similarity transformation used to obtain the different terms in the low energy effective Hamiltonian (Eq. 2). We then describe the generalized Gutzwiller projection for obtaining the projected Hilbert space on which the low energy effective Hamiltonian acts, along with the details of Gutzwiller factors which renormalize the various couplings in the low energy Hamiltonian when the projection is implemented approximately.

We solve the model in Eq. 1, in the limit $U \sim \Delta \gg t, t'$. In this limit and at half-filling, holons are energetically expensive on the $A$ sites (with onsite potential $-\frac{\Delta}{2}$) and doublons are expensive on the $B$ sites (with onsite potential $\frac{\Delta}{2}$); i.e., in the low energy subspace $h_A$ and $d_B$ are constrained to be zero. We do a generalized similarity transformation on this Hamiltonian, $\tilde{H} = e^{-iS}He^{iS}$, such that all first and second neighbour hopping processes connecting the low energy sector to the high energy sector of the Hilbert space are eliminated. The similarity operator of this transformation is $S = -\frac{i}{U+\Delta}(H^+_{t\,A\to B} - H^-_{t\,B\to A}) - \frac{i}{\Delta}(H^0_{t\,A\to B} - H^0_{t\,B\to A}) - \frac{i}{U}(H^+_{t'A\to A} - H^-_{t'A\to A}) - \frac{i}{U}(H^+_{t'B\to B} - H^-_{t'B\to B})$ where

$H^+_{t/t'}$ represents first or second neighbour hopping processes which involve an increase in $h_A$ or $d_B$ by one and $H^-_{t/t'}$ on the other hand represent hopping processes which involve a decrease in $h_A$ or $d_B$ by one. $H^0_t$ processes do not involve a change in $h_A$ and $d_B$. The low energy effective Hamiltonian obtained by this transformation is given in Eq. 2, with $H_0 = \frac{U-\Delta}{2}\sum_i[n_{iA\uparrow}n_{iA\downarrow} + (1-n_{iB\uparrow})(1-n_{iB\downarrow})]$. Further details can be found in [22]. $H_{eff}$ acts on a projected Hilbert space which consists of states $|\Phi\rangle = \mathcal{P}|\Phi_0\rangle$ where the projection operator $\mathcal{P}$ eliminates components with $h_A \geq 1$ or $d_B \geq 1$ from $|\Phi_0\rangle$. We use here the Gutzwiller approximation [13,19,22] to handle the projection, by writing the expectation value of an operator $Q$ in a state $\mathcal{P}|\Phi_0\rangle$ as the product of a Gutzwiller factor $g_Q$ times the expectation value in $|\Phi_0\rangle$ so that $\langle Q\rangle \simeq g_Q\langle Q\rangle_0$. The standard procedure [19] for calculating $g_Q$ has been generalised by us for the case where holons are projected out from one sublattice and doublons from the other [22].

We thus obtain the renormalized effective Hamiltonian with the inter-sublattice kinetic energy $\langle c^\dagger_{iA\sigma}c_{jB\sigma}\rangle \approx g_{t\sigma}\langle c^\dagger_{iA\sigma}c_{jB\sigma}\rangle_0$, and intra-sublattice kinetic energy $\langle c^\dagger_{i\alpha\sigma}c_{j\alpha\sigma}\rangle \approx g_{\alpha\sigma}\langle c^\dagger_{i\alpha\sigma}c_{j\alpha\sigma}\rangle_0$. The inter-sublattice spin correlation $\langle \mathbf{S}_{iA}\cdot\mathbf{S}_{jB}\rangle \approx g_{sAB}\langle \mathbf{S}_{iA}\cdot\mathbf{S}_{jB}\rangle_0$ while the intra-sublattice spin exchange term gets renormalized with a different factor of $g_{s\alpha\alpha}$. The only other dimer term which does not get rescaled under the Gutzwiller projection is

$$H_d = -\frac{t^2}{\Delta}\sum_{<ij>,\sigma}[(1-n_{iA\bar\sigma})(1-n_{jB}) + (n_{iA}-1)n_{jB\bar\sigma}], \tag{3}$$

as it consists of only density operators [19,22].

Then we have the important trimer terms:

$$\begin{aligned}
H_{tr} = &-\frac{t^2}{\Delta}\sum_{<ijk>,\sigma}[g_{A\sigma}c^\dagger_{kA\sigma}n_{jB\bar\sigma}c_{iA\sigma} + g_2 c_{iA\bar\sigma}c^\dagger_{jB\bar\sigma}c_{jB\sigma}c^\dagger_{kA\sigma}] \\
&-\frac{t^2}{\Delta}\sum_{<jil>,\sigma}[g_{B\sigma}c_{lB\sigma}(1-n_{iA\bar\sigma})c^\dagger_{jB\sigma} + g_2 c_{lB\sigma}c^\dagger_{iA\sigma}c_{iA\bar\sigma}c^\dagger_{jB\bar\sigma}] \\
&+\frac{tt'(U+\Delta)}{2U\Delta}\sum_{<kj>,<<ik>>\sigma}\Big[g_{t\sigma}c^\dagger_{iA\sigma}(1-n_{kA\bar\sigma})c_{jB\sigma} - g_{t\sigma}c^\dagger_{jA\sigma}n_{kB\bar\sigma}c_{iB\sigma} \\
&+ g_{AAB\sigma}c^\dagger_{iA\sigma}c^\dagger_{kA\bar\sigma}c_{kA\sigma}c_{jB\bar\sigma} + g_{BBA\sigma}c^\dagger_{jA\sigma}c^\dagger_{kB\bar\sigma}c_{kB\sigma}c_{iB\bar\sigma}\Big] + h.c. \tag{4}
\end{aligned}$$

The various Gutzwiller factors involved (see [22] for details) are as follows:

- $g_{A\sigma} = 2\delta/(1+\delta+\sigma m_A)$, $g_{B\sigma} = 2\delta/(1+\delta-\sigma m_B)$ and $g_{t\sigma} = \sqrt{g_{A\sigma}g_{B\sigma}}$ ;

- $g_{s\alpha_1\alpha_2} = 4/\sqrt{((1+\delta)^2 - m^2_{\alpha_1})((1+\delta)^2 - m^2_{\alpha_2})}$, and $g_2 = \delta g_{sAB}$ ;

- $g_{\alpha_1\alpha_1\alpha_2\sigma} = 4\delta/\sqrt{((1+\delta)^2 - m^2_{\alpha_1})(1+\delta+\sigma m_{\alpha_1})(1+\delta+\sigma m_{\alpha_2})}$ .

**Superconducting order parameter $\Phi_{d/s}$:**
The SC correlation function is the two particle reduced density matrix defined by $F_{\gamma_1\gamma_2}(\mathbf{r}_i-\mathbf{r}_j) = \langle B^\dagger_{i\gamma_1}B_{j\gamma_2}\rangle$ where $B^\dagger_{i\gamma}$, defined above, creates a singlet on the bond $(i, i+\gamma)$. The SC order parameter $\Phi_{d/s}$ is defined in terms of the off-diagonal long-range order in this correlation $F_{\gamma_1,\gamma_2}(\mathbf{r}_i-\mathbf{r}_j) \to \langle B^\dagger_{i\gamma_1}\rangle\langle B_{j\gamma_2}\rangle = \Phi_{\gamma_1}\Phi_{\gamma_2}$ as $|\mathbf{r}_i-\mathbf{r}_j| \to \infty$. Since $F_{\gamma_1\gamma_2}(\mathbf{r}_i-\mathbf{r}_j)$ also corresponds to hopping of two electrons from $(j, j+\gamma_2)$ to sites $(i, i+\gamma_1)$, in the projected wavefunction scheme it scales just like the product of two hopping terms such that $F_{\gamma_1\gamma_2} \approx g_{A\uparrow}g_{B\downarrow}F^0_{\gamma_1\gamma_2}$. Hence the rescaled form of the superconducting order parameter is $\Phi_{d/s} \approx \sqrt{g_{A\uparrow}g_{B\downarrow}}\Phi^0_{d/s}$ where

$\Phi^0_{d/s} \equiv \Delta_{d/s}$ is the order parameter calculated in the unprojected wavefunction of the low energy effective Hamiltonian in Eq. 2.

**Spectral Functions and Density of States:**

In the paper we also discuss the single particle density of states (DOS) and the spectral functions. In the Gutzwiller projection method, the Green's function is rescaled with the appropriate Gutzwiller factor such that $G_{\alpha\sigma}(k,\omega) = g_{\alpha\sigma}G^0_{\alpha\sigma}(k,\omega)$ where $G^0_{\alpha\sigma}(k,\omega)$ is calculated in the unprojected basis. Here $\alpha$ represents the sublattice A or B and $\sigma$ is the spin index. The spectral function, $A_{\alpha\sigma}(k,\omega)$ which is imaginary part of the Green's function also get rescaled with the same Gutzwiller factors.

The single particle density of states is defined as, $\rho_{\alpha\sigma}(\omega) = \sum_k A_{\alpha\sigma}(k,\omega)$. The results presented in the paper are for the single particle density of states (DOS) in the up spin and down spin channels, defined as $\rho_\sigma(\omega) = (\rho_{A\sigma}(\omega) + \rho_{B\sigma}(\omega))/2$. The zero temperature momentum distribution function, which helps in identifying whether a Fermi pocket is an electron pocket or a hole pocket can also be obtained from the spectral function using $n_\sigma(k) = \int_{-\infty}^0 d\omega A_\sigma(k,\omega)$.

# B   Details of renormalized mean field theory (RMFT)

The mean field quadratic Hamiltonian can be written as

$$
\begin{aligned}
H_{MF} = \sum_{k,\sigma} & h_{1A\sigma}(k)c^\dagger_{kA\sigma}c_{kA\sigma} + h_{1B\sigma}(k)c^\dagger_{kB\sigma}c_{kB\sigma} + h_{2\sigma}(k)[c^\dagger_{kA\sigma}c_{kB\sigma} + h.c.] \\
& + h_3(k)[c^\dagger_{kA\uparrow}c^\dagger_{-kB\downarrow} + h.c.] - h_4(k)[c^\dagger_{kB\uparrow}c^\dagger_{-kA\downarrow} + h.c.].
\end{aligned}
\tag{5}
$$

Here $h_{1A\sigma}(k) = T^0_{A\sigma}(k) + T^1_{A\sigma}(k)$ and $h_{1B\sigma}(k) = T^0_{B\sigma}(k) - T^1_{B\sigma}(k)$ with

$$
\begin{aligned}
T^0_{\alpha\sigma}(k) = & -\frac{U-\Delta}{4}\sigma m_\alpha + \frac{t^2}{\Delta}\left[g_{\alpha\sigma}\gamma_{k2}\sigma\frac{m_{\bar\alpha}}{2} - g_{\bar\alpha\bar\sigma}(d\chi_{\bar\alpha\bar\alpha\bar\sigma} + 4^d C_2\chi'_{\bar\alpha\bar\alpha\bar\sigma})\right] + \frac{t^2}{U+\Delta}dg_{sAB}\sigma m_{\bar\alpha} \\
& - t'\gamma_{k3}g_{\alpha\sigma} + \frac{4t'^2}{U}{}^d C_2 g_{s\alpha\alpha}\sigma m_\alpha - \frac{t'^2}{2U}\left[g_{s\alpha\alpha}\chi'_{\alpha\alpha\sigma} + 2g_{s\alpha\alpha}\chi'_{\alpha\alpha\bar\sigma} - \chi'_{\alpha\alpha\sigma}\right]\gamma_{k3} \\
& - \frac{tt'(U+\Delta)}{2U\Delta}\left[16^d C_2 g_{t\bar\sigma}\sum_{i=1}^d \chi^{(i)}_{AB\bar\sigma} + 4dG(\alpha)\chi^{(1)}_{AB\bar\sigma}\gamma_{k3}\right] - \mu \,.
\end{aligned}
\tag{6}
$$

Here $G(A) = g_{AAB\sigma}$ and $G(B) = g_{BBA\bar\sigma}$.

$$
T^1_{\alpha\sigma}(k) = \frac{U-\Delta}{4}(1+\delta) - \frac{t^2}{\Delta}\left[2d(1-2\delta) + g_{\alpha\sigma}\gamma_{k2}\frac{(1-\delta)}{2}\right] + \frac{t^2}{U+\Delta}d(1-\delta) + \frac{4t'^2}{U}{}^d C_2(1-\delta),
\tag{7}
$$

$$
\begin{aligned}
h_{2\sigma}(k) = & \left[-tg_{t\sigma} - \frac{t^2}{\Delta}\left(-2\chi^{(1)}_{AB\sigma} + 2(2d-1)g_2\chi^{(1)}_{AB\bar\sigma}\right) - \frac{t^2}{2(U+\Delta)}g_{sAB}\chi^{(1)}_{AB\sigma} - \frac{t^2}{U+\Delta}g_{sAB}\chi^{(1)}_{AB\bar\sigma}\right. \\
& \left. - \frac{t^2}{2(U+\Delta)}\chi^{(1)}_{AB\sigma} + \frac{tt'(U+\Delta)}{2U\Delta}\left(g_{t\sigma}\frac{m_A+m_B}{2}\gamma_{k3} - 2^d C_2 g_{AAB\bar\sigma}\chi'_{AA\bar\sigma} - 2^d C_2 g_{BBA\sigma}\chi'_{BB\bar\sigma}\right)\right]\gamma_{k1}.
\end{aligned}
\tag{8}
$$

The anomalous components are given by $h_3(k) = -T^\pm_\downarrow(\cos(k_x) \pm \cos(k_y))$ and $h_4(k) = T^\pm_\uparrow(\cos(k_x) \pm \cos(k_y))$ where the + sign corresponds to the extended s-wave case

and the $-$ sign corresponds to the d-wave case. Furthermore,

$$T_\sigma^- = \left[\left\{\frac{2t^2}{\Delta} - \frac{2t^2}{U+\Delta}\left(-\frac{3g_{sAB}}{4} + \frac{1}{4}\right) - \frac{t^2}{\Delta}\left(g_{A\bar\sigma} + g_{B\sigma}\right) - \frac{2t^2}{\Delta}g_2\right\}\Delta_d\right], \tag{9}$$

$$T_\sigma^+ = \left[\left\{\frac{2t^2}{\Delta} - \frac{2t^2}{U+\Delta}\left(-\frac{3g_{sAB}}{4} + \frac{1}{4}\right) + \frac{3t^2}{\Delta}\left(g_{A\bar\sigma} + g_{B\sigma}\right) + \frac{6t^2}{\Delta}g_2\right\}\Delta_s\right]. \tag{10}$$

Here, $\gamma_{k1} = 2\sum_i \cos(k_i)$, $\gamma_{k2} = 2\sum_i \cos(2k_i) + 4\sum_{\substack{i,j \\ i\neq j}}[\cos(k_i + k_j) + \cos(k_i - k_j)]$ and

$\gamma_{k3} = 2\sum_{\substack{i,j \\ i\neq j}}[\cos(k_i + k_j) + \cos(k_i - k_j)]$ where $i,j$ can take x,y or x,y,z values depending upon

whether it is square or cubic lattice. Also, $d$ refers to the number of dimensions in the above Hamiltonian. If $\alpha = A$, then $\bar\alpha = B$ and vice-versa.

In the spin-symmetric SC case, we used a generalized Bogoliubov transformation to solve the mean field Hamiltonian described above, which results in a diagonal Hamiltonian $\mathcal{H} = \sum_k E_1(k)f_{1k}^\dagger f_{1k} - E_1(k)f_{2k}f_{2k}^\dagger + E_2(k)f_{3k}^\dagger f_{3k} - E_2(k)f_{4k}f_{4k}^\dagger + const$. The pairing amplitude obeys the following self consistent equation

$$\Delta_{s,d} = \frac{1}{N}\sum_k (v_{1k}u_{3k} + v_{2k}u_{4k})(\cos(k_x) \pm \cos(k_y)). \tag{11}$$

Here $(u_{1k}, u_{3k}, -v_{1k}, -v_{3k})$ is the eigenvector corresponding to eigenvalue $E_1(k)$ and $(u_{2k}, u_{4k}, -v_{2k}, -v_{4k})$ is the eigenvector corresponding to the eigenvalue $E_2(k)$. The rescaled pairing amplitude, which is the product of coupling strengths and $\Delta_{s,d}$ is given by $h_4(k)$ which decides the gap in the single particle density of states in the SC phase. Thus, the $gap_{(s,d)}$, which gives peak to peak seperation in the single particle density of states, obeys the following equation

$$gap_d = 2\left[\left\{\frac{2t^2}{\Delta} - \frac{2t^2}{U+\Delta}\left(-\frac{3g_{sAB}}{4} + \frac{1}{4}\right) - \frac{t^2}{\Delta}\left(g_A + g_B\right) - \frac{2t^2}{\Delta}g_2\right\}\Delta_d\right], \tag{12}$$

$$gap_s = \frac{1}{2}\left[\left\{\frac{2t^2}{\Delta} - \frac{2t^2}{U+\Delta}\left(-\frac{3g_{sAB}}{4} + \frac{1}{4}\right) + \frac{3t^2}{\Delta}\left(g_A + g_B\right) + \frac{6t^2}{\Delta}g_2\right\}\Delta_s\right]. \tag{13}$$

The spin symmetric spectral function obtained from the sublattice average retarded Green's function, $\frac{1}{2}\sum_\alpha G_{i\alpha,j\alpha}(t)$ where $G_{i\alpha,j\alpha}(t) = -i\theta(t)\langle\{c_{i\alpha}(t), c_{j\alpha}^\dagger(0)\}\rangle$ in this case is

$$A(k,\omega) = \frac{1}{2}[(g_A v_{1k}^2 + g_B v_{3k}^2)\delta(\omega - E_1(k)) + (g_A v_{2k}^2 + g_B v_{4k}^2)\delta(\omega - E_2(k))$$
$$+ (g_A u_{1k}^2 + g_B u_{3k}^2)\delta(\omega + E_1(k)) + (g_A u_{2k}^2 + g_B u_{4k}^2)\delta(\omega + E_2(k))]. \tag{14}$$

Summing of spectral function over the Brillouin zone gives the single particle density of states shown in the manuscript $\rho(\omega) = \sum_k A(k,\omega)$.

**Non-pairing phase**: In the non-pairing phase, the sublattice Green's functions are given by

$$G_{AA\sigma}(k,\omega) = \frac{\alpha_{k\sigma}^2}{\omega - E_{1\sigma}(k) + i\eta} + \frac{\beta_{k\sigma}^2}{\omega - E_{2\sigma}(k) + i\eta},$$
$$G_{BB\sigma}(k,\omega) = \frac{\beta_{k\sigma}^2}{\omega - E_{1\sigma}(k) + i\eta} + \frac{\alpha_{k\sigma}^2}{\omega - E_{2\sigma}(k) + i\eta}. \tag{15}$$

Here, $(\alpha_{k\sigma}, -\beta_{k\sigma})$ is the eigenvector corresponding to the eigenvalue $E_{1\sigma}(k)$ and $(\beta_{k\sigma}, \alpha_{k\sigma})$ is the eigenvector corresponding to the other eigenvalue $E_{2\sigma}(k)$. This gives the following expression of the sublattice averaged spectral function

$$A_\sigma(k,w) = \frac{1}{2}[(g_{A\sigma}\alpha_{k\sigma}^2 + g_{B\sigma}\beta_{k\sigma}^2)\delta(\omega - E_{1\sigma}) + (g_{A\sigma}\beta_{k\sigma}^2 + +g_{B\sigma}\alpha_{k\sigma}^2)\delta(\omega - E_{2\sigma})]. \tag{16}$$

In order to get the low energy spectral functions, we integrate $A_\sigma(k, \omega)$ over a small $\omega$ range such that $|\omega| \le (0.01 - 0.02)t$.

The Gutzwiller factors $g_{A\sigma} = 2\delta/(1 + \delta + \sigma m_A)$ and $g_{B\sigma} = 2\delta/(1 + \delta - \sigma m_B)$ are different in the ferrimagnetic phase but equal in the para and the AF phases. Thus from the normalization condition of the eigenfunctions, one can see that in the para metal phase $A_\sigma(k, \omega)$ does not vary over the Fermi pockets where it is non-zero. Even in the ferri-metal phase, the quasiparticle weight changes only by a small amount over the Fermi pockets, though the quasiparticle weights for the sublattice resolved spectral functions change significantly over the Fermi pockets.

The sublattice magnetization is determined self-consistently by the following equation

$$m_A = \frac{1}{N} \sum_k [\alpha_{k\uparrow}^2 \Theta(-E_{1\uparrow}(k)) - \alpha_{k\downarrow}^2 \Theta(-E_{1\downarrow}(k))] + [\beta_{k\uparrow}^2 \Theta(-E_{2\uparrow}(k)) - \beta_{k\downarrow}^2 \Theta(-E_{2\downarrow}(k))]. \quad (17)$$

$m_B$ is obtained by interchanging $\alpha_{k\sigma}$ and $\beta_{k\sigma}$ in the above expression.

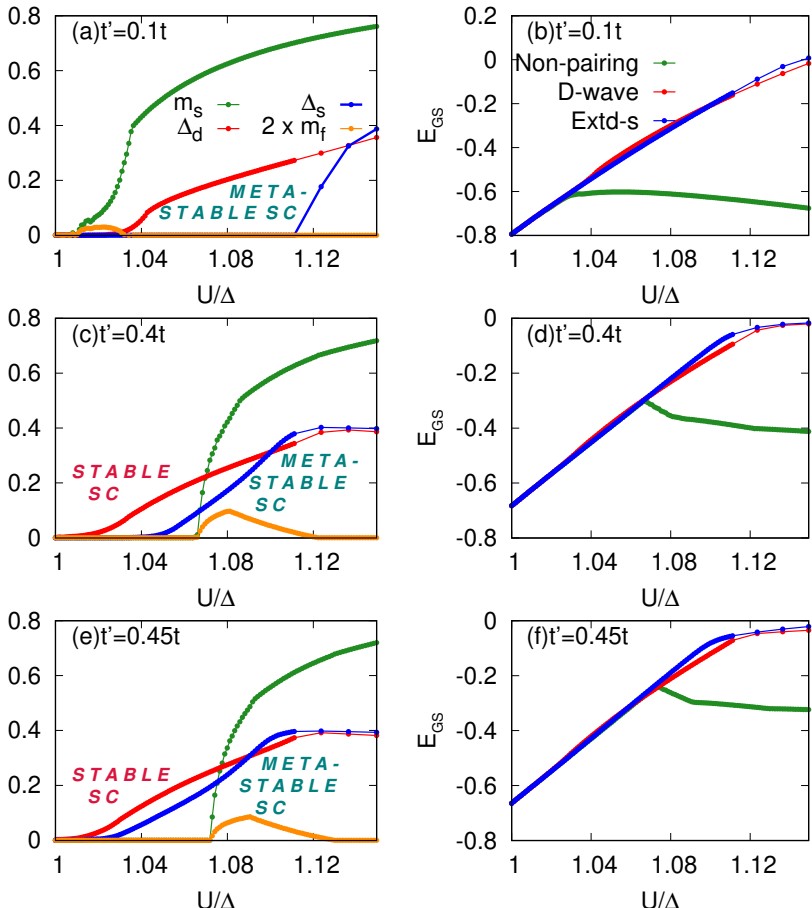

Figure 7: **Order parameters and the ground state energy**. Left panels show various mean fields, namely, the staggered magnetization $m_s$, uniform magnetization $m_f$, d-wave pairing amplitude $\Delta_d$ and the extended s-wave pairing amplitude $\Delta_s$ as functions of $U/\Delta$ for different values of $t'$ at $U = 10t$ for the 2d square lattice. Right panels show the ground state energies for the d-wave SC phase, extended s-wave SC phase and the non-superconducting phase where only magnetic order is allowed, as functions of $U/\Delta$.

# C Competing order-parameters and ground state energy comparison

We solve the effective low energy Hamiltonian using three different versions of renormalized mean field theory (RMFT), the first which allows for superconductivity but not magnetic order, the second which allows for the magnetic order but not superconductivity, and the third which allows for both, along with various other mean fields, as discussed in Section 3 of the paper. When we compare the results from the first two calculations, we find that there is a significantly broad regime of parameters over which the SC and magnetic orders both exist and compete with each other. In order to determine the true nature of the ground state in this parameter regime, we compare the ground state energies of the different RMFT solutions.

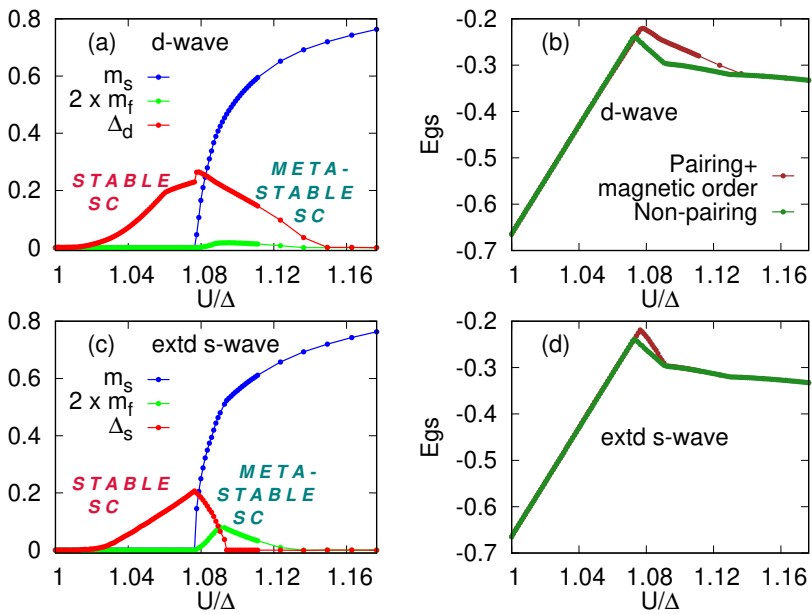

Figure 8: **Comparison of different renormalized mean field theories**. Top left panel shows several mean fields obtained from the third solution of the RMFT where both SC pairing and magnetic order are allowed, namely, the staggered magnetization $m_s$, uniform magnetization $m_f$, and the d-wave pairing amplitude $\Delta_d$ as functions of $U/\Delta$ for $t' = 0.45t$ and $U = 10t$. Top right panel shows the ground state energy of the non-superconducting phase where only magnetic order is allowed and the energy for the third solution as functions of $U/\Delta$. Note that the phase with both orders coexisting is only a metastable phase. Lower panels show similar results for the extended s-wave SC order.

As shown in Fig. 7 , even for small values of $t'$, the SC pairing amplitudes, in both the pairing channels studied, turn on but the magnetic transition precedes the transition into the SC phase. Once the magnetic order turns on, the ground state energy of the non-superconducting solution becomes lower than that of both the SC phases studied as shown in the right panels of Fig. 7. Thus for $t' < 0.1t$ there is no stable SC phase, as shown in Fig. 2[e] of the main paper. For larger values of $t'$, as $U/\Delta$ increases superconductivity turns on before the magnetic order sets in. There continues to be a solution of the RMFT with pairing amplitudes, in either of the symmetry channels, non zero even in the magnetically ordered regime, but the non-superconducting magnetically ordered solution is lower in energy here. Thus the pure SC

phase is a stable phase only before the magnetic transition point.

There is a third scenario possible where one can do a RMFT allowing for non-zero values of both SC and magnetic order parameters along with other mean fields. Before the magnetic order turns on, this theory is consistent with the spin-symmetric Bogoliubov theory described above. After the magnetic order sets in, differences between the two calculations become visible. In the third calculation, the SC order coexists with the ferrimagnetic order for a range of parameters as shown in Fig. 8 though the pairing amplitudes decrease with increasing $U/\Delta$. Comparing the energy of this phase with that of the ferrimagnetic metal phase, which was found to be the stabler phase by comparing the energies in the first two calculations in this regime, we find that the coexistence phase is also a metastable phase, and the system actually stabilizes into the ferrimagnetic metallic phase as shown in Fig. 2 of the paper.

## D   Details of the Phase-diagram

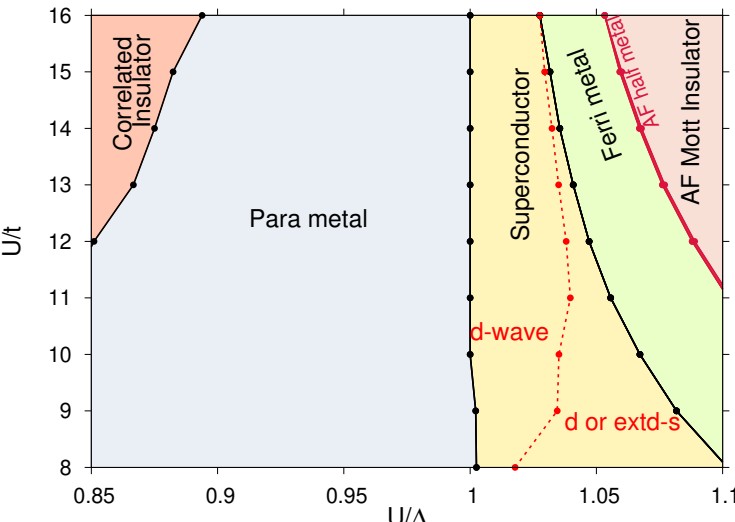

Figure 9: **Phase diagram in $U/t - U/\Delta$ plane**. Phase diagram of the half-filled IHM on a 2d square lattice in $U/t - U/\Delta$ plane for $t' = 0.4t$. Note that the SC phase always turns on for $U \sim \Delta$ irrespective of the value of $U/t$ within the range of validity of the calculation. As $U/t$ increases, the range of $U/\Delta$ over which both the s-wave and the d-wave SC phases are viable solutions and almost degenerate shrinks rapidly while the range of $U/\Delta$ over which only the d-wave SC phase is stable reduces rather slowly.

Earlier in this paper we have shown and discussed the phase-diagrams for the IHM on a 2d square lattice for a fixed value of $U/t$. Fig. 2[e] shows the phase diagram in $t'/t - U/\Delta$ plane for a fixed $U$ and Fig. 1 shows a section of this phase diagram for $t' = 0.4t$. In order to understand how the different phases and the phase boundaries between them evolve with varying $U$, here we show in Fig. 9 the phase diagram in $U/t - U/\Delta$ plane for a fixed $t'/t$. As is clear from the figure, superconductivity always turns on for $U \sim \Delta$ irrespective of the value of $U/t$ though with increase in $U/t$, the range of $U/\Delta$ over which both pairing symmetries are almost degenerate solutions shrinks rapidly such that eventually, for large enough values of $U/t$, the system has only a d-wave SC phase.

All the results presented so far in the paper are for the 2d square lattice. We would like to

emphasize that within the renormalized mean field theory the phase diagram is qualitatively similar for higher dimensional systems as well. This is clear from Fig. 10 which shows the phase diagram for a cubic lattice.

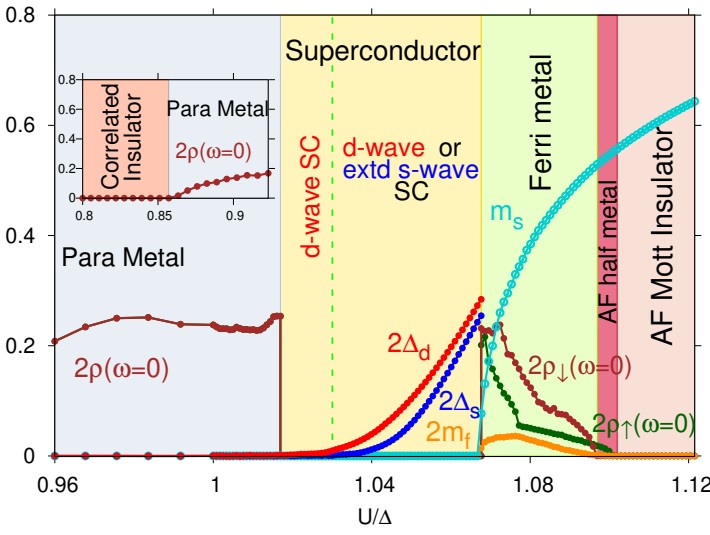

Figure 10: **Phase diagram for cubic lattice**. Phase diagram of the half-filled IHM on a 3d cubic lattice for $U = 12t$ and $t' = 0.35t$. Note that the phase diagram obtained for cubic lattice is qualitatively similar to the one obtained for a 2d square lattice.

## E  Band dispersion and nature of Fermi pockets

Fig. 11 shows the band dispersion $E_{n\sigma}(k)$ for both the bands on paths along high symmetry directions in the BZ. In the AF half-metal phase, the down spin channel has small hole pockets around $\mathbf{K}'$ and tiny electron pockets around $\mathbf{K}$. In the ferrimagnetic metal phase, the down spin band $E_{1\downarrow}(k)$ crosses the Fermi energy around the $\mathbf{K}'$ points resulting in small hole pockets and $E_{2\uparrow}(k)$ crosses the Fermi energy near the $\mathbf{K}$ points resulting in small electron pockets. In the paramagnetic metal phase, $E_1(k)$ crosses the Fermi energy around the $\mathbf{K}'$ points resulting in hole pockets and $E_2(k)$ crosses the Fermi level around $\mathbf{K}$ resulting in electron pockets, where, because of the spin symmetry, we have suppressed the spin indices.

Finally, we show the low energy spectral function $A_\sigma(k, \omega \sim 0)$ for the AF half-metal phase (see Fig. 12), which is fully consistent with the band-dispersions shown above. The up-spin channel is gapped while $A_\downarrow(k, \omega \sim 0)$ has tiny electron pockets at the $\mathbf{K}$ points and hole pockets at the $\mathbf{K}'$ points in the BZ.

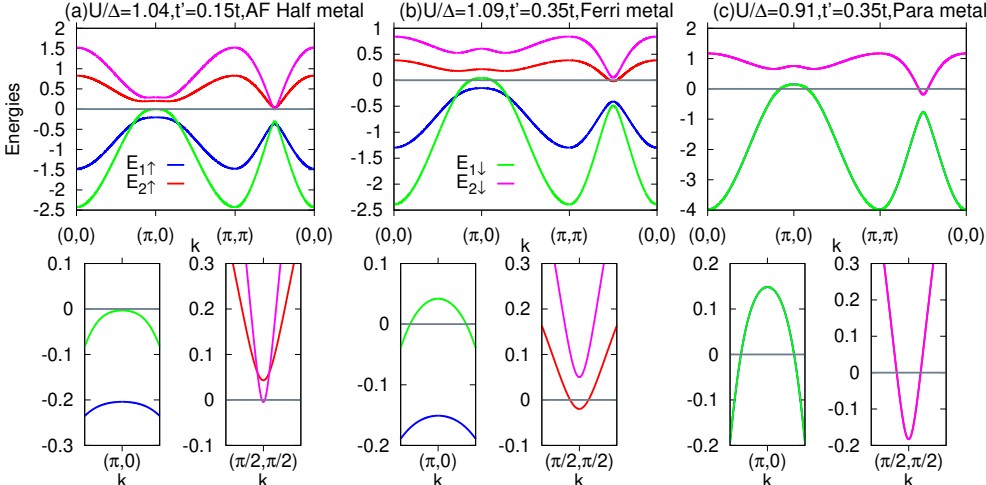

Figure 11: **Band Dispersion**. Band dispersion $E_{n\sigma}(k)$ on paths along high symmetry directions in the BZ. Panel (a) shows bands in the AF half-metal phase where both the down spin bands cross the Fermi level near the **K** and **K'** points while the up spin bands are fully gapped. Panel (b) shows bands in the ferrimagnetic metal phase, where one down-spin band crosses the Fermi level near the **K'** points while one up-spin band crosses the Fermi level near the **K** point and the other two bands are gapped. Panel (c) shows bands in the paramagnetic metal phase where there is a spin symmetry and all the bands cross the Fermi level. The lower panels zoom in close to the band crossing at the Fermi energy.

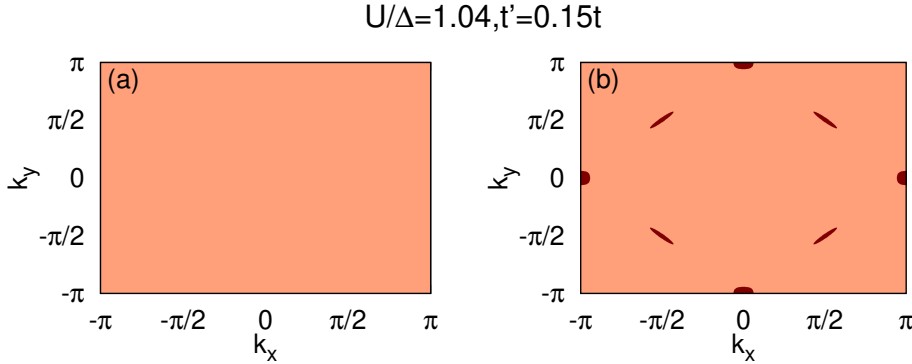

Figure 12: **Spectral functions in the AF half-metal phase**. Spin resolved low energy spectral function $A_\sigma(k, \omega \sim 0)$ (integrated over $|\omega| \leq 0.01t$) in the AF half-metal phase. Left (right) panel shows the spectral function for the up-spin (down-spin) channel.

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
