# Peer review of "Unconventional superconductivity in a strongly correlated band-insulator without doping"

_SciPost Physics, doi:SciPost Phys. Core 4, 009 (2021)_

## Round 1 · Referee Report · Anonymous (Referee 1) · 2021-2-24

Strengths

1) New interesting ideas on d-wave and extended s-wave SC at half-filling within a frustrated systems through t' only.

Weaknesses

1) The separation of high and low-energy degrees of freedom is an important approximation in this context, which might lead to weak results.

Report

In the manuscript, the authors study the existence of unconventional superconductivity (SC) by using the ionic Hubbard model without doping. While being a band-insulator, a large staggered potential, comparable to the Hubbard interaction, closes the band-gap and the system shows a metallic state even at half-filling, as also shown in Ref. (24). Contrary to the pure Hubbard model, double occupancy is not energetically prohibited allowing for charge fluctuations. By using the Gutzwiller projection, the authors are able to derive a low-energy t-t'-J-J' effective Hamiltonian, which is then solved by using the renormalized mean-field theory (RMFT) for the magnetic and superconducting gaps.

The main results is shown in Fig.(1), where the authors found a superconducting state surrounded by paramagnetic metal and ferri metal states. As the system approaches the SC phase from the two sides, the increase of the electron pockets in the quasi-particle spectrum lead to pairing mechanisms at the Fermi surface.

The main part of the article is well written and the results discussed extensively. The physical ideas are still important to warrant a publication. Hence, I recommend the publication of the manuscript once the authors addressed the comments below.

Requested changes

1) It is not clear how spin fluctuations are treated and how they generate a sizable pairing gap in optics of the RMFT. I think that the paper would benefit from a better presentation of the RMFT in general, by reporting the gap equations and the magnetic correlators.

2) The authors comment that their calculations estimate the superconducting critical temperature in cuprates. One should carefully consider that the absence of doping and a too large staggered potential might lead to a physically different situation when considering the critical temperatures. Also in the context of the spectral function, that is significantly different from the observed in cuprates. Indeed, as shown in Fig.(3), all the quasi-particles have the same spectral weight along the Fermi-surface.

3) The value of t' is positive (with respect to t), while usually a negative t' is used. The difference lead to the presence of hole-pockets at k=(\pi/2,\pi/2) instead of el-pockets. How would the system behave with a negative t'?

  • validity: good
  • significance: good
  • originality: high
  • clarity: high
  • formatting: good
  • grammar: excellent

Author:  Arti Garg  on 2021-03-15  [id 1309]

(in reply to Report 1 on 2021-02-24)

We thank the referee for positive assessment of our work. We have provided detailed response to all the queries raised by the referee in "referee_reply.pdf" file.

Attachment:

referee_reply.pdf

Anonymous on 2021-03-22  [id 1321]

(in reply to Arti Garg on 2021-03-15 [id 1309])

I thank the authors for convincingly addressing my comments. I therefore recommend the manuscript for publication.

---

## Round 2 · Author Response

Dear Prof. Andergassen,

We are resubmitting our manuscript to Scipost Physics. The referee has given a strong positive assessment of our work and has also recommended it for publication provided we reply to the queries raised. We have prepared a detailed response to all the queries the referee has raised and also made relevant changes in the manuscript. We have uploaded the response to the referee ("referee_reply.pdf" ) and the list of changes in the manuscript is given below. We hope that the manuscript will be accepted for publication soon.

Best regards,
Arti Garg
(On behalf of all authors)

---

## Round 2 · List of Changes

List of changes in the manuscript

1) Added Appendix B following referee’s comment (1). Also modified section 3 on page 4 accordingly.

2) Added a few lines about sign of t’ and the related particle-hole transformation in the Section 2 on page 3 below equation (1).

3) Edited last line in the third paragraph on page 7 about the estimation of $T_c$ keeping the referee’s comment in mind.

4) In the paragraph below Fig.4 on page 9, added a few sentences clarifying the issue of spectral weight being the same over the Fermi pockets.

5) Combined earlier Appendices (C,D) into Appendix D

6) Combined Appendices E and F into Appendix E.

---

## Editorial Decision

published